# Immunothrombosis in COVID-19: Implications of Neutrophil Extracellular Traps

**DOI:** 10.3390/biom11050694

**Published:** 2021-05-06

**Authors:** Brandon Bautista-Becerril, Rebeca Campi-Caballero, Samuel Sevilla-Fuentes, Laura M. Hernández-Regino, Alejandro Hanono, Al Flores-Bustamante, Julieta González-Flores, Carlos A. García-Ávila, Arnoldo Aquino-Gálvez, Manuel Castillejos-López, Armida Juárez-Cisneros, Angel Camarena

**Affiliations:** 1Laboratorio HLA, Instituto Nacional de Enfermedades Respiratorias Ismael Cosío Villegas, Mexico City 14080, Mexico; brandon.bautistab@gmail.com (B.B.-B.); armida1109@yahoo.com.mx (A.J.-C.); 2Programa MEDICI, Carrera Médico Cirujano, FES Iztacala, Universidad Nacional Autónoma de México, Mexico City 54090, Mexico; beckycampi@yahoo.com.mx (R.C.-C.); glezf.julie@gmail.com (J.G.-F.); 3Laboratorio de Neuropsicofarmacología, Departamento de Farmacología, Facultad de Medicina, Universidad Nacional Autónoma de México, Mexico City 04510, Mexico; 4Departamento de Infectología, Hospital General de México Eduardo Liceaga, Mexico City 06720, Mexico; samuelsevilla2007@gmail.com; 5Escuela Nacional de Ciencias Biológicas, Programa de Posgrado, Instituto Politécnico Nacional, Mexico City 11340, Mexico; lauramoon@comunidad.unam.mx (L.M.H.-R.); carlosgar0796@gmail.com (C.A.G.-Á.); 6Facultad de Ciencias de la Salud, Universidad Anáhuac México Norte, Mexico City 52786, Mexico; ahanono95@gmail.com; 7Laboratorio de Farmacología, Instituto Nacional de Pediatría, Mexico City 04530, Mexico; albb.18@gmail.com; 8Laboratorio de Biología Molecular, Departamento de Fibrosis Pulmonar, Instituto Nacional de Enfermedades Respiratorias Ismael Cosío Villegas, Mexico City 14080, Mexico; araquiga@yahoo.com.mx; 9Departamento de Epidemiología Hospitalaria e Infectología, Instituto Nacional de Enfermedades Respiratorias Ismael Cosío Villegas, Mexico City 14080, Mexico; mcastillejos@gmail.com

**Keywords:** immunothrombosis, COVID-19, neutrophil extracellular traps, CID, SARS-CoV-2

## Abstract

SARS-CoV-2 is a member of the family of coronaviruses associated with severe outbreaks of respiratory diseases in recent decades and is the causative agent of the COVID-19 pandemic. The recognition by and activation of the innate immune response recruits neutrophils, which, through their different mechanisms of action, form extracellular neutrophil traps, playing a role in infection control and trapping viral, bacterial, and fungal etiological agents. However, in patients with COVID-19, activation at the vascular level, combined with other cells and inflammatory mediators, leads to thrombotic events and disseminated intravascular coagulation, thus leading to a series of clinical manifestations in cerebrovascular, cardiac, pulmonary, and kidney disease while promoting severe disease and mortality. Previous studies of hospitalized patients with COVID-19 have shown that elevated levels of markers specific for NETs, such as free DNA, MPO, and H3Cit, are strongly associated with the total neutrophil count; with acute phase reactants that include CRP, D-dimer, lactate dehydrogenase, and interleukin secretion; and with an increased risk of severe COVID-19. This study analyzed the interactions between NETs and the activation pathways involved in immunothrombotic processes in patients with COVID-19.

## 1. Introduction

Coronavirus disease 2019 (COVID-19) is an emerging condition caused by the SARS-CoV-2 virus, and since December 2019, it has represented a serious public health problem that has compromised health systems worldwide [1]. Many patients develop mild forms of the illness, but several studies have shown that some patients progress to a severe acute respiratory syndrome, sepsis, coagulopathy, and multiorgan failure—mainly elderly subjects with certain comorbidities such as diabetes mellitus, hypertension, and obesity [2,3,4]. Thrombosis has been reported in a great number of patients, representing a significant cause of death in patients with COVID-19. Neutrophil extracellular traps (NETs) are structures produced by neutrophils to confine infections. These are formed by chromatin meshes, antimicrobial peptides, and enzymes that, when released into the extracellular space, immobilize microorganisms and facilitate their death [5,6,7]. Research has shown that diverse pathogens including viruses induce NET formation. Though beneficial for the host’s defense, the collateral damage from the sustained formation of NETs can trigger a cascade of inflammatory reactions. Such reactions can result in the destruction of surrounding tissues, promote microthrombosis, and cause permanent damage to the organs of the pulmonary and cardiovascular systems, among others [6,7,8]. Elevated serum levels of NETs have been reported in hospitalized patients with COVID-19, suggesting that these structures may be key in the pathophysiological process of the disease and have an association with the patients’ prognosis [9]. Therefore, in this review, the role of NETs as a triggering mechanism for thrombotic processes in patients with COVID-19 is analyzed.

## 2. SARS-CoV-2 Overview

Coronaviruses (CoVs) belong to the Coronaviridae family; this includes four genera: *Alphacoronaviruses*, *Betacoronaviruses*, *Gammacoronaviruses*, and *Deltacoronaviruses*. *Alphacoronaviruses* and *Betacoronaviruses* (βCoVs) have been known to infect mammals. SARS-CoV-2 is a βCoV with a positive, single-stranded RNA genome. It has a spherical morphology and specific elements that enable its replication [1,10,11].

Over the last several decades, three major outbreaks involving βCoV for which severe acute respiratory syndrome coronavirus (SARS-CoV-1) and Middle East respiratory syndrome coronavirus (MERS-CoV) were responsible have been reported. On December 2019, a series of pneumonia cases were reported in Wuhan City, Hubei Province, in China, and a new coronavirus (SARS-CoV-2) was identified as the etiological agent, whose genome presents a 79.6% similarity to the SARS-CoV-1 sequence [1,10,11,12,13,14].

## 3. Mechanism of SARS-CoV-2 Infection

It was recently explained that the first step in SARS-CoV-2 virus entry is the binding of the viral trimeric spike (S) protein to the angiotensin-converting enzyme 2 (ACE2) receptor [2,10,11,15]. ACE2 cleaves angiotensin II and thereby counteracts ACE1’s vasoconstrictive activity but is also known to be a receptor for SARS-CoV-1. The S protein is a type I viral entry protein that is processed in two domains, S1 and S2. The first binds to ACE2, while S2 mediates the viral membrane’s fusion with the target cells [2,10,12,16].

A SARS-CoV-2 infection triggers an inflammatory response characterized by an increased production of IL-6, IL-8, and IFN-γ and the activation of the nuclear factor enhancing the kappa light chains of activated B cells (NF-kB), jun N-terminal kinase (JNK), and phosphatidylinositol 3-kinase (PI3K) pathways, which are necessary for its permanence [2,11,17,18].

## 4. Epidemiology

The World Health Organization (WHO) named the disease caused by SARS-CoV-2, COVID-19. Despite China’s efforts to halt COVID-19’s transmission, the infection spread throughout Asia, and cases were reported in Thailand, Japan, and South Korea by January 2020 [19]. Within three months of its discovery, the infection had spread to at least 114 countries, causing more than 4000 deaths [20]. On 11 March 2020, the WHO announced the outbreak of COVID-19 as a pandemic [21]. By 16 February 2021, there were a total of 108.2 million globally confirmed cases, resulting in 2.3 million deaths and a total of 2.7 million new cases per week, with a mortality rate varying from 6.1 to 111.2 in different countries; in December 2020, the mortality rate ranged from 0.3 to 13.1. [22,23].

## 5. Neutrophils and NETS

The innate immune response is the frontline defense against pathogens mediated through the most abundant population of circulating leukocytes, the neutrophils [24,25,26]. These are granular leukocytes about 12–15 μm in diameter, with nuclei segmented into three-to-five lobes. Adults produce more than 1 × 10^11^ neutrophils every day [27,28], normally comprising more than 60% of the nucleated cells in the bone marrow and bloodstream [29] and having a typical circulating average life of six-to-eight hours in blood and then migrating through the tissues for two-to-three days [25,26,27].

Neutrophil function can be summarized in four simple stages: recognition and binding, phagocytosis, destruction, and NET formation. The first stage and most important recognition mechanism in neutrophils occurs through toll-like receptors (TLRs) (which are pattern recognition receptors (PRRs) expressed on their surface), through their endosomes, or within their cytoplasm [24,29,30,31]. These receptors are activated when they recognize structures that are not normally present in the host cells and/or that are shared by multiple pathogens (including viruses) [32]. A variety of TLRs have been specifically identified for different infectious agents, such as types 4, 7, 8, and 9 [25,29,33].

The next stage begins when a particle of a microorganism is engulfed, forming a vacuole, called a phagosome, that fuses with a lysosomal granule containing antimicrobial peptides and enzymes, allowing for the binding of the granules’ content to form a phagolysosome [31,34]. In this cellular compartment, microorganisms are exposed to high concentrations of reactive oxygen species (ROS), and nicotinamide adenine dinucleotide phosphate hydrogen (NADPH) subunits transfer electrons to oxygen to form superoxide anions that, in turn, are catalytically converted to dioxygen and hydrogen peroxide, which are responsible for microbial destruction and NET formation [35].

Due to the changes produced by ROS, neutrophils lose their characteristic lobed architecture, and euchromatin and heterochromatin are homogenized. There is a rupture in their nuclear membranes and a dilution of the cytoplasmic granules, which allow for their contents to be released externally, thus forming NETs [8,35,36]. This process represents an alternative antimicrobial function of neutrophils based on a special form of programmed cell death called NETosis, different from apoptosis and necrosis [28,30,35,37]. Neutrophils initially present autophagy, but if it is inhibited, NET formation is also inhibited and the neutrophils die from apoptosis [38,39].

Though ROS-dependent NETosis is the classical mechanism, there is an early ROS-independent mechanism where neutrophils can release NETs without the activation of the NADPH oxidase complex. The ROS-independent mechanism is called vital NETosis, where NETs are released without cell death, thus keeping their normal functions, including phagocytosis. This vital NETosis occurs quickly and usually within 5–60 min of the cells being stimulated [40,41].

NETs are formed by a mesh of chromatin fibers, antimicrobial peptide granules, and enzymes that are released into the extracellular space to immobilize microorganisms, trapping them and facilitating their death [5]. Bacteria, fungi, viruses, and several protozoan parasites have been implicated in inducing NET formation, a function of neutrophils beneficial for controlling the spread of infectious microorganisms [6].

The contents of NETs have nuclear, granular, or cytoplasmic locations in non-stimulated neutrophils [5,26,28,37,38]. (Table 1). Among the nuclear components, the most abundant are histones. The four histone subtypes—H2A, H2B, H3, and H4—have immune activity and represent 70% of all NET-associated proteins. Ten granular proteins have also been identified; the most important are lactoferrin, gelatinase, elastase, and myeloperoxidase (MPO) [28,30,42,43].

## 6. Biomolecules and Cellular Elements Involved in the Interaction with NETS

### 6.1. Cytokine Storm

It has been demonstrated that COVID-19 patients can develop a cytokine storm, the toxicity of which is potentially fatal, leading to detrimental effects such as capillary leakage, tissue toxicity, edema, coagulopathy, organ failure, and shock [11]. A cytokine storm is induced by the dysregulated activation of a variety of WBCs, including B-lymphocytes, T-lymphocytes, NK cells, macrophages, dendritic cells, neutrophils, and monocytes, as well as epithelial or endothelial cells, which release high amounts of proinflammatory cytokines such as IL-2, IL-6, IL-8, IL-7, IL-1β TNF-α, CRP, ferritin, and D-dimer [2,7,17,44].

Cytokines and inflammatory mediators play important roles in the activation of NETosis, which can be induced by antibodies, immune complexes, cytokines, and chemokines [45]. Once the virus is recognized by the TLRs themselves, circulating monocytes and macrophages are recruited into the alveolar space and activated by mediators such as macrophage colony stimulating factor (M-CSF) produced by T cells, macrophages, endothelial cells, and fibroblasts [46].

Alveolar macrophages (AMs) exhibit two main phenotypes: M1 and M2. M1 macrophages stimulate inflammation by secreting proinflammatory cytokines, such as IL-1β, IL-12, TNF-α, IL-6, and inducible nitric oxide synthase (iNOS). M2 macrophages promote tissue repair due to their anti-inflammatory functions, which are mediated by the release of Th2 cytokines, such as IL-4, IL-10, and IL-13. TNF-α and IL-1β activate neutrophils and induce the overexpression of adhesion molecules, such as intercellular and vascular adhesion molecule 1, in immune and endothelial cells [47]. However, TNF-α, IL-1β, and IL-8 have been reported to play a fundamental role in the triggering of neutrophil oxidative burst and the formation of NETs [48].

Interestingly, cathelicidin (LL-37), which is a cationic peptide synthesized by NETs, can stimulate the chemotaxis of macrophages and promote their differentiation, stimulating the macrophages to continue releasing cytokines. In COVID-19, damage-associated molecular patterns (DAMPs) generated during the lytic viral replication cycle signal through the inflammasome of the NOD-3-like receptor protein (NLRP3), resulting in the processing of the IL-1β precursor molecule, which is then secreted by macrophages. This, in turn, induces the formation of NETs, suggesting an NET–IL-1β–IL8 loop that enhances the production of NETs, IL-8, and IL-1β during SARS-CoV-2 replication, which aggravates tissue damage [47,49,50].

Cytokines and extracellular histones in SARS patients have been found to excessively increase and may contribute to the progression of the disease by inducing cell damage and promoting a procoagulant state at both the pulmonary and systemic levels; all of this is secondary to the activation of the complement system, platelets, and coagulation pathways [51,52,53,54].

### 6.2. Complement System

Over the course of COVID-19, an increase in specific IgM has been observed during the acute phase, followed by an increase in specific IgG in the subsequent phases; these can also produce immunocomplexes, contributing to lung tissue destruction, inflammation, intravascular coagulation, and the activation of the complement system [2,55]. In patients with sepsis, complement proteins decrease by consumption, which suggests an exaggerated immune response secondary to a cytokine storm and increased fibrogenesis [56]. In the pulmonary microvasculature of COVID-19 patients, Magro et al. found deposits of the terminal complement complex C5b-9, the membrane attack complex (MAC), C4d, mannose-binding lecithin (MBL), and MBL-associated serine protease 2 (MASP2). These are factors associated with a sustained systemic activation of the alternative pathway (AP) and the lectin pathway (LP) [57]. It has been documented that SARS-CoV-2 and its S-glycoprotein cause the cleavage of C4 and C2 by the MBL complex along with MASP2 to form C3 convertase (C4bC2a), which, in turn, activates the complement cascade [58].

NETs also activate the complement system. Myeloperoxidase, cathepsin G, and proteinase 3 activate propidine, factor B, and C3—three components of the alternative pathway necessary to induce the complement cascade. Notably, the activation of the complement system has been reported in severe COVID-19 patients. Together, neutrophil infiltration and NET formation drive necroinflammation during all coronavirus infections [59]. The complement system plays an important role because C1 inhibits NET degradation by inhibiting DNase I. In general, there is a positive feedback loop where NETs activate the complement system, which then keeps NET levels elevated by preventing their degradation [60].

Furthermore, experiments in mouse models have revealed that histone–DNA complexes bind type C lectins through the histone moiety and activate macrophages through TLR-9-dependent endosomal responses. This results in the secretion of proinflammatory cytokines such as IL-6 or TNF-α, along with IL-1β—important drivers of the cytokine storm observed during the later stages of COVID-19 [46].

The complement activation product C3a activates platelets, and C5a increases the expression and activity of a potent coagulation initiator, the tissue factor (TF), in both macrophages and the endothelium [61]. Reciprocally, it is possible for factor Xa, thrombin, and factor IXa to trigger the complement cascade by acting as independent C3 and C5 convertases [62]. Given the proposal for the use of C5/C5a antibodies in the treatment of COVID-19 [63,64,65], Gao et al. observed clinical improvement when severe patients were treated with anti-C5a monoclonal antibodies [66].

### 6.3. Platelets

Interactions between platelets and neutrophils are mediated by their surface molecules such as *p*-selectin (CD62P), which binds to the *p*-selectin glycoprotein ligand 1 (PSGL-1) on the neutrophils’ surfaces, inducing platelet activation [67]. Circulating platelet–neutrophil complexes occur in a diverse range of inflammatory disorders and infections, such as unstable angina, myocardial infarction, cardiopulmonary bypass sequelae, thrombosis, and sepsis. These platelet–leukocyte interactions are important processes that enhance the local accumulation of neutrophils, monocytes, and, potentially, other inflammatory cell types in inflamed or injured tissues. These interactions also further amplify both platelet and neutrophil activation, and they position platelets as a key regulator of neutrophil activation [68].

Alternatively, platelets can be activated upon binding constituents of the platelet membrane glycoprotein (GPIb-IX-V) complex, such as integrin αIIbβ3, and upon interacting with integrin αMβ2 in neutrophils, either directly or through fibrinogen, which acts as a bridge molecule; this increases the thrombus size, enabling the formation of intravascular fibrin [69,70].

Cathepsin G and neutrophil elastase (NE), present in NETs, can directly modify platelet function to enhance fibrin formation by activating platelets through the protease-activated receptor 4 (PAR4) and thrombin with a high specificity. Their cleavage leads to the complete activation of platelets, including remodeling, the activation of glycoprotein IIb–IIIa, and (to a lesser extent) the activation of plasma zymogens such as coagulation factors X and V [71,72].

It has also been demonstrated that citrullinated histone H3 (H3cit), present in NETs, interacts with the von Willebrand factor (vWF) and promotes the development of a thrombus rich in erythrocytes and platelets because it stimulates the secretion of Weibel–Palade bodies in the latter, exacerbating the production of vWF [73,74]. It has been observed that the binding of histones to platelets is non-saturable, which means that histones interact with the plasma membrane directly or through their receptors, increasing platelet activation. In this way, histones reassociate with platelets, providing new binding sites, enhancing their activation, favoring their consumption, and activating platelet adhesion molecules and thrombin-dependent fibrin formation [75,76]. It has also been proven that platelet *p*-selectin prepares neutrophils for NETosis [77]. This suggests a vicious cycle of microvascular thrombosis, where platelets accumulate on the neutrophils’ surface and, by doing so, activate NETosis, subsequently causing platelet aggregation. Studies in mice have found that, three hours after histone administration, lungs presented tissue damage, hemorrhaging, platelet-rich intravascular thrombi, and neutrophil infiltration. Additionally, lungs showed that platelets are removed from blood circulation in the first few minutes after histone infusion, and platelets along with histones are deposited in the lungs [75,78].

In response to the preceding findings, it was found that heparin binds to histones and then prevents their binding to platelets [79]. However, surprisingly, in a study by Elaskalani et al. DNase and heparin-treated NETs were still able to aggregate washed platelets, and they also induced the expression of *p*-selectin and activated αIIbβ3 in the same proportion as non-treated NETs. Though DNase and heparin can destabilize the structure of NETs, they do not inhibit NET-induced platelet aggregation [80].

Since platelet integrins include αIIbβ3 (GPIIb–IIIa) and other integrins such as αVβ3, α2β1, α5β1, and α6β1, Monti et al. used different cell lines with a low expression of αIIbβ3, as well as the expression of many other integrins such as αVβ3; by using antibodies against α5β1 and αvβ3 in treated and non-treated NETs with DNase 1, they achieved an inhibition of adhesion with the antibodies similar to that obtained with DNase 1 treatment, though DNAse 1 and the anti-integrin antibody treatment together almost completely blocked cell adhesion, confirming that both DNA and fibronectin were relevant in determining cell attachment to NETs [81,82].

In another study, Monti et al. showed that an excess of the cyclic RGD peptide inhibited cell adhesion to NETs by competing with fibronectin within NETs. The maximal reduction of such adhesion was similar to that obtained by DNase 1 treatment causing DNA degradation. The findings indicated that RGD-binding integrins may play a major role in the cell adhesion of different carcinoma cells to NETs, since high levels of α5β1, αvβ3, and αvβ5 in cells enhance adhesion to NETs, whereas a low expression of α5β1 prevents cell adhesion to NETs [82]. Other mechanisms involved in platelet and cell adhesion have recently been studied [83,84].

Regarding SARS-CoV-1, which broke out in China in 2002, it has been suggested that the combination of infection and mechanical ventilation leads to endothelial injury, which triggers platelet aggregation and overactivation in the lung [85]. It was found that 40–50% of patients infected with SARS-CoV-1 developed thrombocytopenia and diffuse alveolar injury, which entrapped megakaryocytes and hindered their release of platelets [85,86,87,88,89]. It is possible that SARS-CoV-2 causes thrombocytopenia in a similar way. A study regarding COVID-19-associated coagulopathy reported that 71.4% of the deceased met the International Society on Thrombosis and Haemostasis (ISTH) criteria for Disseminated Intravascular Coagulation (DIC) [44].

In DIC, due to the consumption of coagulation factors and platelets, the lengthening of the PT and the decrease in the platelet count and fibrinogen predominate. In COVID-19, fibrinogen is elevated, and the platelet count initially remains normal and subsequently tends to the lowest values within the normal range, but the severe thrombocytopenia seen in DIC has not been observed [90,91].

Yang et al. conducted a meta-analysis on 1476 COVID-19 patients and found a higher mortality rate in those with thrombocytopenia; nonetheless, mortality decreased as platelets increased, suggesting a reduction in thrombosis, that platelets are no longer consumed, and that an increase in platelets may be an indicator of clinical improvement [92].

A study in China found a rapid increase in the platelet count prior to the onset of the severe form of the disease in 113 patients with COVID-19. The peak level was reached on days 11–15 after the initial clinical manifestations; thereafter, the platelet count rapidly decreased, correlating with the severity of the disease [93]. In a meta-analysis that included nine studies with a total of 1779 COVID-19 patients, it was reported that the platelet count was significantly lower in patients with severe COVID-19. Furthermore, in a stratified analysis of the population under study, it was determined that a low platelet count was associated with up to five times the risk of developing severe forms of the disease, once again demonstrating that the platelet count is closely related to the progression of the disease [94].

Based on findings obtained using infectious disease models, we now understand that platelets can participate in the host immune response via several mechanisms. At sites of sterile injury, platelets quickly alter the microenvironment, initiating inflammatory responses and helping to establish the ensuing tissue repair response [68]. Since platelets normally circulate in large numbers and are often present early at sites of tissue injury, platelets may serve an important role as immune sentinels, helping to initiate and drive the innate immune response to infection [68].

### 6.4. Coagulation Cascade

It has been observed that histones can upregulate TFIII, which activates the extrinsic pathway and factors X and V, responsible for converting fibrinogen into the active fibrin, in addition, these can bind to the phosphatidylserine contained in cells, activating proteins of the contact pathway (which involves factor XI and factor XII), initiating the intrinsic coagulation pathway and increasing thrombin generation even in the absence of platelets, thus promoting intravascular thrombogenesis [95,96,97]. On the other hand, DNA and histones H3 and H4 downregulate thrombomodulin, which leads to an increase in the procoagulant phenotype [95,98].

Thrombin, factor VIIa, and factor Xa, among other coagulation factors, are activated and regulated through protease activated receptors (PARs), such as serine, elastase, and cathepsin G—all components of NETs. PAR activation also promotes the endothelial expression of TF and the mobilization of Weibel–Palade bodies (which store and release vWF) as positive feedback to platelet activation [87,99,100,101].

A study conducted by Semararo et al. showed that histone H4 contributed to the activation of the coagulation cascade through inorganic polyphosphate (PolyP), which acts as platelet activator and contributes to the activation of coagulation factors V and XI, this describes a new dynamic axis called NET–platelet–thrombin, which promotes intravascular coagulation and microvascular dysfunction in sepsis [102,103,104]. Di Micco et al. studied 67 COVID-19 patients whose values of certain coagulation factors were compared, and it was found that patients with COVID-19 and SARS had statistically significantly higher levels of fibrinogen (747 mg/dL), associated with a more severe course of the disease [95].

Following a study conducted by Song, J.C. et al. recommended, among other actions, the evaluation of the coagulation function in COVID-19 patients with routine coagulation tests. Therefore, they proposed the evaluation of the extrinsic pathway through PT or the international normalized ratio (INR), as well as the intrinsic pathway through the activated partial thromboplastin time (aPTT). The evaluation of the common pathway by using the thrombin time (TT) and fibrinogen was also recommended [105] because patients who died of severe COVID-19 developed DIC around day 4, and a significant worsening of all these parameters was also reported between days 10 and 14 [106].

However, a study by Fogarty et al. with a total of 83 patients showed that there was no significant difference in PT values at the time of hospital admission, but there was a significant difference in the values of the ICU patients [107]. Tang et al. observed a steady increase in D-dimer and PT during the first seven days after hospital admission in Chinese patients diagnosed with COVID-19 who did not survive, in contrast to survivors, whose PT and D-dimer levels remained consistent within the normal range or slightly increased during this period [44]. Free thrombin in the circulation, uncontrolled by natural anticoagulants, can activate platelets and stimulate fibrinolysis. At critical stages of COVID-19, the levels of fibrin-related markers, such as D-dimer and fibrin degradation product (FDP), were markedly elevated in all the patients who died, suggesting a common activation of coagulation and a secondary hyperfibrinolytic condition in these patients [44,107]. Other studies have shown that elevated D-dimer levels, together with an increased sequential organ failure assessment (SOFA) score and an advanced age with or without risk factors, may be useful in determining the prognosis of patients in early stages of SARS-CoV-2 infection [108].

Moreover, Yu Zuo et al. found elevated levels of free DNA, MPO-DNA, and H3Cit, which are specific markers of NETs, in hospitalized COVID-19 patients. It is noteworthy that free DNA is strongly associated with the total neutrophil count and with acute-phase reactants, which include CRP, D-dimer, and lactate dehydrogenase (LDH). In this study, mechanically ventilated patients showed a higher elevation of free DNA and MPO-DNA but no elevation of H3Cit. Ultimately, COVID-19 patients’ sera triggered NETosis from control neutrophils in vitro [9,109]. Middleton et al. observed numerous citrullinated histone H3+ and MPO+ PMNs and rare lattices of extracellular DNA decorated with citrullinated histone H3 and MPO in autopsy lung specimens. They also observed significant increases in plasma NET levels in non-intubated COVID-19 patients and endotracheally intubated COVID-19 patients compared with healthy donors and convalescent patients (*p* = 0001). Plasma NET levels were significantly higher in COVID-19 non-survivors than in COVID-19 survivors (*p* = 0.0004) [109]. These data suggest a correlation between serum NET levels and the severity of COVID-19 (Figure 1).

Finally, thrombocytopenia and elevated D-dimer levels result from the overactivation of the coagulation cascade and from platelet consumption. Viral infections induce a systemic inflammatory response and lead to an imbalance between the procoagulant and anticoagulant homeostatic mechanisms. Multiple pathogenic processes, including endothelial dysfunction, the elevation of vWF, TLR activation, and TF pathway activation, are involved [101].

### 6.5. Biological Markers in COVID-19

SARS-CoV-2-mediated lung infection leads to the cellular release of multiple cytokines that induce the recruitment of immune cells as a direct action on resident cells and as a result of the release of cytokines into the bloodstream. The resulting mechanism leads to lung injury and edema, resulting in interstitial pneumonia and acute respiratory distress syndrome (ARDS). Preliminary studies have shown statistically significant differences in biological markers when comparing mild and severe forms of the disease due to infection with SARS-CoV-2 (Table 2).

There are two molecules related to NETs that are associated with the severity of SARS-CoV-2 lung infection; cell-free DNA and MPO-DNA have been found to be elevated in patients with the severe form of the disease [109,110]. On the other hand, Cit-H3, a main component of NETs, has been found to be increased in both forms of the disease with no significant difference [111,112,113].

Regarding the factors associated with coagulation, many studies have shown that D-dimer is increased in the severe form of the disease compared to the mild form. It is even strongly associated with mortality as an isolated factor [114]. Regarding platelet count, it has been found that in mild COVID-19, platelets tend to normal counts, while in severe COVID-19, there is a noticeable decrease due to their destruction by the immune system and the platelet aggregation in the lungs, resulting in microthrombi and platelet consumption, which is attributed to the dysregulation of the underlying immunity, SOCS 1 mutations, and other mechanisms including molecular mimicry, cryptic antigen expression, and epitope propagation [115]. No specific interaction between the platelet count and the severity of the disease has been found; however, a reduction below 150 × 10^3^ has been associated with greater severity in the ICU (taking the SOFA range into account) [116].

Elevated ferritin levels have been associated with severe COVID-19 and, in combination with clinical and laboratory data, can be considered for disease stratification [117,118]. In the case of anaphylatoxins, elevated levels of C3a and C5a have been reported in severe disease compared to healthy controls [119,120]. IL-1b has been found to be increased during the course of the disease; however, no significant differences in IL-1b were found when comparing the severe and mild forms. On the other hand, the levels of IL-6 and IL-10 were higher in the severe form than in the mild form [121,122].

The role of white blood cells in COVID-19 has been evaluated, and a strong association has been found between decreased lymphocyte counts and worse disease progression, as well as increased mortality [54]. In addition, it has been observed that the neutrophil/lymphocyte ratio (NLR) in severe patients is higher than it is in mild patients; an NLR value greater than 4.8 has been described as a predictor of severity in patients with COVID-19, with a sensitivity and specificity of 78% [123].

Recent studies have pointed to genetic factors as the main determinant of susceptibility to infection by SARS-CoV-2 or the severity of the disease since the components of the immune response to the virus seem to be related to interindividual variation in the HLA, TLR7, and IRF7 genes, among others, while the genes related to the binding of the virus to the ACE2 receptor and its entry largely determine the susceptibility of each individual. These studies could facilitate the development of precision medicine strategies for the prevention and timely care of severe COVID-19 [124,125].

## 7. Immunothrombosis, NETS, and COVID

Over the last few years, several studies have explored the role of NETs in thrombotic diseases, in which they infiltrate and spread through the vessel wall, occluding arteries, veins, and microscopic vessels when formed intravascularly, thus leading to lung, heart, and kidney damage [126,127,128,129]. There are recent reports of severe complications and deaths in COVID-19 patients due to several coagulation-related mechanisms. A study in the Chinese population of Wuhan found heart injury in 20% of 416 hospitalized patients diagnosed with COVID-19, while in another study conducted in the Netherlands, 38% of 184 patients with COVID-19 had some coagulopathy. Finally, an analysis of clinical data showed that over 70% of patients who died from COVID-19 presented clear signs of DIC compared to only 0.6% of COVID-19 survivors who presented this complication [3,4]. According to the data generated on thrombotic diseases, the most frequent complications in patients with COVID-19 include acute myocardial infarction (AMI), cerebrovascular disease (CVD), venous thromboembolism (VTE), sepsis, and DIC [3,130,131,132].

### 7.1. Myocardial Infarction

In myocardial infarction, the formation, stabilization, and development of a coronary thrombus are crucial processes that determine the clinical outcome resulting from a plaque rupture. Neutrophils have demonstrated a potential role in these processes through the formation of NETs, which have been observed in a significant number of coronary thrombi. They have even been associated with the infarction size, which suggests their capacity for spreading thrombosis [133,134]. Farkas et al. described a direct correlation of DNA and H3Cit concentrations with the age of patients. Since histones confer mechanical and lytic resistance to fibrin, clot stabilization could be greater in patients of advanced age, meaning that such patients would show poor responses to conventional lytic treatment [135].

The circulating cytokines released during severe systemic inflammatory stress, such as those observed in COVID-19 patients, may lead to the instability and rupture of atherosclerotic plaques. A cohort study conducted by Shi et al. in 416 hospitalized patients with COVID-19 showed that 19.7% of the patients had evidence of myocardial injury manifested by increased levels of troponin I (TnI) [136,137]. This finding was also observed by Guo et al. in 27.8% of 187 patients infected with SARS-CoV-2, in which myocardial injury was determined by increased levels of troponin T (TnT) [136,138]. COVID-19 patients with concomitant cardiovascular disease are at a higher risk of severe complications from the disease. Moreover, it has been observed that COVID-19 patients without pre-existing cardiovascular disease are at risk of developing cardiovascular injury, such as acute myocardial injury, myocarditis, arrhythmias, and VTE [136,139,140]. The understanding of the mechanisms leading to cardiovascular disorders is expanding, as these disorders overlap with regulatory pathways of immune function [140]. All of the above findings suggest an increased risk of developing myocardial injury in COVID-19 patients with concomitant cardiovascular disease.

### 7.2. Cerebrovascular Disease

NETs are present in large quantities in cerebral thrombotic processes, and their content has been associated with the number of times an endovascular procedure device has been introduced in order to restore the permeability of the affected vessel [141,142]. A significant elevation of DNA, nucleosomes, and H3Cit has been observed to be directly correlated with the initial National Institutes of Health Stroke Scale (NIHSS) score, and it is particularly higher in patients with cardioembolic strokes [143]. Additionally, the co-administration of tissue plasminogen activator (tPA) along with DNase accelerates thrombolysis compared to tPA or DNase alone in these same patients, which may reflect a resistance to tPA by NETs [142].

COVID-19 has been associated with neurological symptoms and complications, including CVD, due to the hypercoagulable conditions that are associated with the infection. In a retrospective study involving 214 hospitalized patients with COVID-19 in China, 5.7% of the patients with severe disease presented acute CVD [144]. Earlier this year, up to one-third of patients with COVID-19 were reported to have at least one neurological symptom, such as headaches, seizures, and coma. The pathogenesis of the neurological complications in COVID-19 is complex and involves a set of interactions between viral properties, tissue susceptibility, and the host immune response [130,131].

### 7.3. Venous Thromboembolism

Though the aggregation of NETs during the organization phase of venous thrombogenesis has been described, a close association between NETs and lysophosphatidic acid (LPA) has recently been observed in intrapulmonary thrombi, as has LPA showing a significant elevation in the plasma of patients with pulmonary thromboembolism (PTE). LPA is a bioactive phospholipid, released by activated platelets, that induces the release of NETs. These may, in turn, activate further platelets to release more LPA and perpetuate the LPA–NET–platelet positive feedback mechanism [145,146]. In a study by Jimenez et al. histone levels were strongly associated with the degree of VTE and with mortality, as well as with C-reactive protein (CRP) and circulating leukocytes [147]. Similarly, in other studies on thrombotic diseases, a significant acceleration of intrapulmonary thrombus lysis by a combination of DNase and tPA has also been observed [146].

According to a study by Klok et al. in a group of 184 patients hospitalized in the intensive care unit (ICU) with COVID-19 pneumonia, the most frequent thrombotic complication was pulmonary thromboembolism (31%). Additionally, the presence of thrombotic venous disease was observed in 27% of cases and arterial thrombotic events were observed in 3.7% of cases by using computed tomography and/or ultrasound [148]. In Spain, an increased risk of VTE in patients with COVID-19 pneumonia admitted to the ICU has been reported. Whether COVID-19 increases the risk of VTE in non-ICU wards remains unknown [149].

### 7.4. NETs and Possible Therapeutic Intervention Points

NETosis inhibitors are in clinical development for combating the toxic activities of NETs. Among the studied substances, we found the neonatal NET inhibitory factor (nNIF), which was found for the first time in umbilical cord blood and has been shown to inhibit key terminal events in the formation of NETs, including the activity of peptidyl arginine deiminase 4 (PAD4), the citrullination of neutrophil nuclear histones, and nuclear decondensation. nNIF, PAD4, and the serine protease protection peptide (CRISPP) have been shown to effectively inhibit the formation of NETs in human and murine studies. In a study by Yost et al. when additional nNIF-related peptides (NRP) were administered to lipopolysaccharide (LPS)-challenged mice, they provided an early survival advantage in this model and reduced mortality in a polymicrobial sepsis model, suggesting that NRPs have potential as anti-inflammatory therapies in specific syndromes in which the formation of NETs contributes to acute or progressive pathological inflammation [150]. On the other hand, recombinant human DNase, marketed as Pulmozyme (Dornase alfa) by Genentech, is a highly purified solution of recombinant human deoxyribonuclease I (rhDNase). This is an enzyme that selectively cleaves DNA and has been used to hydrolyze the DNA from NETs that are present in the sputum/mucus of patients with cystic fibrosis, which reduces the viscosity in the lungs, thus promoting the elimination of secretions [151]. Studies have demonstrated the destructive effect of DNase on DNA nucleoprotein and immune complexes, showing a reduction in NETosis with less neutrophil infiltration and reducing the inflammatory response. The effect of rhDNase on NETs in patients with sepsis showed that early and concurrent treatment with DNase I and antibiotics resulted in better survival, a reduction in bacteremia, and protection against organ dysfunction in septic conditions, suggesting a possible combination therapy for controlling NETosis [152]. On the other hand, sivelestat (a selective neutrophil elastase inhibitor) was approved for treating ARDS in Japan and South Korea, and a new generation of powerful NETosis inhibitors, including lonodelestat (POL6014), alvelestat, CHF6333, and elafin, have all undergone phase I testing. As a result, it may be possible to accelerate their development as part of the directed therapies for the treatment of COVID-19 [7].

## 8. Conclusions

The COVID-19 pandemic has generated a complex scenario for health systems all around the world and has had substantial social and cultural impacts on individuals’ lives.

The spread has been accelerating such that programs for specific treatments and for vaccine design have been implemented to protect the population. At present, although vaccines against this pathogen have become a reality, it is important to keep focusing on researching and identifying the associated molecules that modify the prognosis of the disease, as well as their use as therapeutic targets.

Neutrophils are being extensively studied in coronavirus disease, as they are the first line defense of the innate immune response against microorganisms. Elevated levels of these cells have been associated with the formation of extracellular neutrophil traps. NETs are one of the host’s immunity mechanisms for eliminating microorganisms, mainly viruses and bacteria. Several studies have shown the effectiveness of these structures in combating viral agents. However, their excessive production is detrimental because it can lead to an exaggerated immune response that causes a cytokine storm that leads to serious complications such as impaired cardiorespiratory and neurological function, eventually leading to death.

In individuals with COVID-19, elevated levels of specific NETosis markers that, when associated with other biomolecules and cellular elements, confer an increased risk of severe COVID-19 have been observed. In this review, we propose a model for the interactions of these NETs in the activation and propagation of the different thrombosis mechanisms, such as the coagulation cascade, the complement system, and platelets. However, further studies are needed in patients with COVID-19 to demonstrate the correlation between NET levels and the severity of the disease. This would allow for proposing them as therapeutic targets and, therefore, for reducing inflammation, thrombotic processes, and mortality from the disease.

## Figures and Tables

**Figure 1 biomolecules-11-00694-f001:**
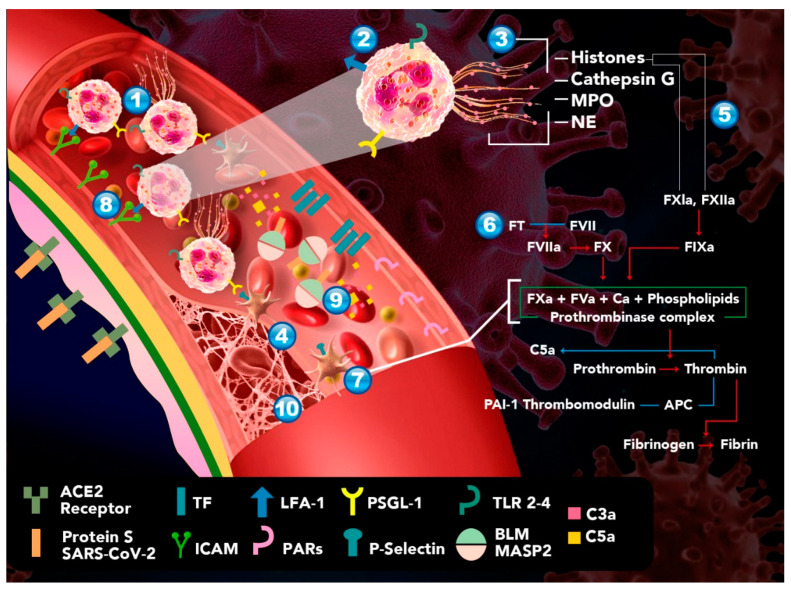
Proposed model of hypercoagulability in patients with COVID-19. **1**. Increased secretion of inflammatory interleukins and chemokines recruits neutrophils to the site of infection. **2**. Unable to eliminate the virus by activation of TLR-2, and -4, the neutrophils release their extracellular traps by NETosis. **3**. NETs are rich in histones, free DNA, and MPO acting as DAMPs, amplifying the inflammatory response and leading to endothelial damage. **4**. The previous step leads to platelet aggregation, clot formation, and stabilization. **5**. Histones H3 and H4 present in NETs activate the intrinsic coagulation pathway through their interaction with FXI and XII, and they downregulate thrombomodulin, inducing a procoagulant state. **6**. Endothelial injury leads to the activation of the extrinsic pathway by the expression of TFIII, which binds to FVII, triggering the coagulation cascade. **7**. Thrombin, FXa, and the TFIII–FVII complex interact with protease activated receptors (PARs), causing platelet activation and aggregation with the subsequent release of their granular content, such as *p*-selectin. **8**. *p*-selectin favors the activation and migration of additional PMNs that easily bind to the endothelium through adhesion molecules—leukocyte function antigen-1 (LFA-1) and intercellular adhesion molecule-1 (ICAM-1). **9**. Meanwhile, the complement system activates thrombin by binding C3a and C5a. **10**. Perpetuating thrombus formation.

**Table 1 biomolecules-11-00694-t001:** Location of NET contents in non-stimulated neutrophils.

Location in Non-Stimulated Neutrophil	Content
Nucleus and Mitochondria	DNA
Nucleus	Histones (70%)
	H1/H2A/H2B/H3/H4
Cytoskeleton	Actin
	Myosin-9
	Calprotectin
Granules	Elastases
	Myeloperoxidases
	Matrix metalloprotease 8 and 9
	Lysozymes
	Cathepsin G and C
	Pentraxin 3
	Ficolin I
	Cathelicidin (LL-37)
	Lactoferrin
	Gelatinases
Peroxisomes	Catalases

**Table 2 biomolecules-11-00694-t002:** Biological markers in patients with mild and severe COVID-19.

Biomolecules and Cell Elements	Markers	Mild COVID-19	Severe COVID-19	*p*
NETs	Cit-H3	↑	↑	>0.05
	Cell-free DNA	↑	↑↑	<0.0001
	MPO-DNA	↑	↑↑	<0.05
Coagulation factors	Fibrinogen	NV	↑	0.784
	DD	↑	↑↑	<0.001
	Platelets	NV	↓	<0.001
Anaphylatoxins	C3a	NV	↑	<0.05
	C5a	NV	↑	<0.05
Proinflammatory markers	IL-1b	↑	↑	0.82
	IL-6	↑	↑↑	0.01
	IL-10	↑	↑↑	0.022
	Ferritin	<400 ng/dL	>400 ng/dL	<0.001
Cells	Lymphocytes	↓ or NV	↓↓	0.001
	Neutrophils	↑ or NV	↑	0.08
	NLR	-	>4.8	0.04
Genetic factors	*HLA*	HLA-B*15:03	HLA-B*46:01	-
	*TLR7*	-	g.12905756_12905759del g.12906010G>T	0.002
	*IRF7*	-	Deletion	<0.001

Cit-H3: 3-citrullinated histone; MPO-DNA: myeloperoxidase–DNA complex; DD: D-dimer; C3a–C5a: anaphylatoxins; IL-1b: interleukin 1b; IL-6: interleukin 6; IL-10: interleukin 10; NLR: neutrophil lymphocyte ratio; HLA: human leukocyte antigen; TLR7: toll-like receptor 7; IRF7: interferon regulatory factor 7. ↑: higher than health control; ↑↑: higher than mild COVID-19; ↓: less than health control; ↓↓: less than mild COVID-19; NV: normal value.

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
