# Peer review of "Immunothrombosis in COVID-19: Implications of Neutrophil Extracellular Traps"

_biomolecules, 2021, doi:10.3390/biom11050694_

Round 1

Reviewer 1 Report

In the current manuscript, the authors review the function of neutrophil extracellular traps (NETs)-induced thrombosis in COVID-19. Since thrombus formation and exacerbation of COVID-19 is of great concern now, the manuscript will provide important information to researchers in a wide range of fields. The authors provide an overview of SARS-CoV-2 infection, host responses, and NETs formation and present a model for thrombus formation in COVID-19. However, it is unfortunate that there is a lack of discussion on the mechanisms of cytokine-induced regulation of NETs formation and therapeutic strategies targeting NETs in COVID-19.

Furthermore, the English text is not correctly written, making it difficult to understand the contents properly. Also, the text and figures are not well-coordinated, making it difficult for a wide range of readers to understand. For example, it is questionable whether Figure 1 is necessary for the topic. The insertion into the section on "receptors and signaling pathways" on page.2 is also incongruous. Also, the connection between Figure 2 and the text is not clear. In addition, there are errors in the text, figures, and references, which need to be re-proofread carefully. Note that it is difficult for the reviewer to determine if the citations are correct because several Spanish papers are cited in references.

Author Response

Title: “Immunothrombosis in COVID-19, implications of Neutrophil Extracellular Traps”

Reviewer 1. Responses.

Point A:  In the current manuscript, the authors review the function of neutrophil extracellular traps (NETs)-induced thrombosis in COVID-19. Since thrombus formation and exacerbation of COVID-19 is of great concern now, the manuscript will provide important information to researchers in a wide range of fields. The authors provide an overview of SARS-CoV-2 infection, host responses, and NETs formation and present a model for thrombus formation in COVID-19. However, it is unfortunate that there is a lack of discussion on the mechanisms of cytokine-induced regulation of NETs formation and therapeutic strategies targeting NETs in COVID-19.

Response.  We appreciate the suggestion of including a section discussing the role of cytokines in the regulation of NETs formation and another section focused on reviewing therapeutic strategies targeting NETs as therapeutic targets in COVID-19. The first topic is described on page 4 in the Biomolecules and cellular elements involved in NETs interaction section. While the second topic was included on page 13 in the section NETs and possible therapeutic intervention targets.

Point B. Furthermore, the English text is not correctly written, making it difficult to understand the contents properly. Also, the text and figures are not well-coordinated, making it difficult for a wide range of readers to understand. For example, it is questionable whether Figure 1 is necessary for the topic. The insertion into the section on "receptors and signaling pathways" on page.2 is also incongruous. Also, the connection between Figure 2 and the text is not clear. In addition, there are errors in the text, figures, and references, which need to be re-proofread carefully. Note that it is difficult for the reviewer to determine if the citations are correct because several Spanish papers are cited in references.

Point 1.The English text is not correctly written, making it difficult to understand the contents properly.

Response 1: The English text was sent to the MDPI editorial for review. The language edition certificate is attached.

Point 2. The text and figures are not well-coordinated, making it difficult for a wide range of readers to understand. For example, it is questionable whether Figure 1 is necessary for the topic. The insertion into the section on "receptors and signaling pathways" on page.2 is also incongruous.

Response 2: Thanks for your suggestion. The inclusion of Figure 1 in the section "receptors and signaling pathways" is incorrect. Furthermore, the virus replication has been extensively described in other publications. In response to your comments, we decided to delete Figure 1.

Point 3.  The connection between Figure 2 and the text is not clear.

Response 3: We appreciate your comment on the figure “Physiopathology model proposed” which was not well coordinated. To make the figure clearer to the reader, numbers were included in the figure and the figure caption. These changes are shown on page 9.

Point 4. There are errors in the text, figures, and references, which need to be re-proofread carefully

Response 4: The text, figures and references were carefully examined and sent for language review to the MDPI editorial.

Point 5. It is difficult for the reviewer to determine if the citations are correct because several Spanish papers are cited in references.

Response 5: Each of the references was rechecked and cited in AMA format. Citations from book chapters were reviewed and modified according to the original source of information. As for articles cited in Spanish, they were replaced by scientific articles cited in English. These changes are shown in the Bibliography section (citations 24-26, 33, and 44).

Comments: According to the modifications made to the text, we decided to review the conclusions. The changes to this section are described on page 14, lines 4 and 19.

Reviewer 2 Report

This article provides an overview of the role of neutrophil exracellular traps (NETs) in promoting immunothrombosis in patients with SARS-CoV-2 infection. The authors report existing evidences of the role of NETs in determining myocardial infarction, cerebrovascular disease and venous thromboembolism in COVID-19 patients. Then they examine the results of studies evaluating the ability of NETs to activate complement system, platelets and coagulation cascade in COVID-19 patients. The review addresses an urgent issue and focuses on main aspects of the problem.

English should be revised.

Authors should avoid repetitions throughout the text.

Abstract should be improved describing the content of the review article and being less generic.

Page 2. Receptors and signaling pathways. The title of this paragraph is not appropriated. It would be better ”Mechanism of SARS CoV-2 infection”

Figure 1 should be mentioned in the paragraph on the mechanism of viral infection because it does not report signaling pathways. In this figure it is not clear what N,S,M and E mean.

Page 4. Neutrophils and NETs.  A ROS-dependent and a ROS-independent mechanism of netosis have been reported. Authors should mention both mechanisms.

Page 4. Neutrophils and NETs. The following sentence is not clear “ Neutrophils initially undergo autophagy, but if this is inhibited so is NETosis and neutrophils will die through apoptosis” This sentence should be reworded.

Page 8. Platelets. “ DNase and heparin-treated NETs were still able to aggregate washed platelets and they also induced the expression of P-selectin and activated αIIbβ3 in the same proportion as non-treated NETs.”  In this respect, it would be helpful to mention a couple of studies reporting the interaction of NETs with RGD binding integrins  (PLoS One. 2017 Feb 6;12(2):e0171362; Int J Mol Sci  2018 Aug 9;19(8). pii: E2350. doi: 10.3390/ijms19082350)

Page 10 Coagulation cascade “Ultimately, COVID-19 patients’ serums triggered NETosis from control neutrophils in-vitro”. Please, change serums with sera. Here it would be appropriate to cite also a study by Middleton EA et al Blood. 2020;136(10):1169-1179.

Author Response

Title: “Immunothrombosis in COVID-19, implications of Neutrophil Extracellular Traps”

Reviewer 2. Responses.

This article provides an overview of the role of neutrophil extracellular traps (NETs) in promoting immunothrombosis in patients with SARS-CoV-2 infection. The authors report existing evidence of the role of NETs in determining myocardial infarction, cerebrovascular disease and venous thromboembolism in COVID-19 patients. Then they examine the results of studies evaluating the ability of NETs to activate complement system, platelets and coagulation cascade in COVID-19 patients. The review addresses an urgent issue and focuses on main aspects of the problem.

Point 1: English should be revised.

Response 1: The English text was sent to the MDPI editorial for review. The language edition certificate is attached.

Point 2: Authors should avoid repetitions throughout the text.

Response 2: The text was carefully revised to avoid confusion and repetition.

Point 3: Abstract should be improved describing the content of the review article and being less generic.

Response 3: The abstract was further revised and includes findings of the association between NETS with other biomolecules and cellular elements, as well as with the severity of SARS-Cov-2 infection. We included a paragraph on page 1.

Point 4: Page 2. Receptors and signaling pathways. The title of this paragraph is not appropriated. It would be better “Mechanism of SARS CoV-2 infection”

Response 4: Thank you for your comment. The title is not appropriate because the signaling pathways are not discussed, therefore we decided to change the title of the paragraph to "Mechanism of SARS CoV-2 infection". The title of the paragraph is shown on page 2.

Point 5: Figure 1 should be mentioned in the paragraph on the mechanism of viral infection because it does not report signaling pathways. In this figure it is not clear what N,S,M and E mean.

Response 5:  We agree that the inclusion of Figure 1 in the receptor and signaling pathways section is incorrect. Furthermore, the replication of the virus has been extensively described in other publications. Following the suggestion of two of the reviewers, we decided to remove figure 1. 

Point 6: Page 4. Neutrophils and NETs.  A ROS-dependent and a ROS-independent mechanism of netosis have been reported. Authors should mention both mechanisms.

Response 6: The classic mechanism of NETOSIS is dependent on ROS, as you point out, another mechanism has been reported that does not require the activation of the NADPH oxidase complex which is called Vital Netosis or ROS Independent. We included a paragraph in the Neutrophils and NETs section (page 3 and line 32).

Point 7: Page 4. Neutrophils and NETs. The following sentence is not clear “Neutrophils initially undergo autophagy, but if this is inhibited so is NETosis and neutrophils will die through apoptosis” This sentence should be reworded.

Response 7: The sentence has been reviewed and as it was very confusing it has been reworded. The sentence is reworded on page 3 and line 29.

Point 8: Page 8. Platelets. “DNase and heparin-treated NETs were still able to aggregate washed platelets and they also induced the expression of P-selectin and activated αIIbβ3 in the same proportion as non-treated NETs.”  In this respect, it would be helpful to mention a couple of studies reporting the interaction of NETs with RGD binding integrins  (PLoS One. 2017 Feb 6;12(2):e0171362; Int J Mol Sci  2018 Aug 9;19(8). pii: E2350. doi: 10.3390/ijms19082350)

Response 8: The articles were reviewed and provided further information in the platelet section, particularly with regard to the study of other mechanisms involved in cell adhesion and NETs. This information is given on page 7 line 3.

Point 9: Page 10 Coagulation cascade “Ultimately, COVID-19 patients’ serums triggered NETosis from control neutrophils in-vitro”. Please, change serums with sera. Here it would be appropriate to cite also a study by Middleton EA et al Blood. 2020;136(10):1169-1179.

Response 9: The suggested change was made and the citation from Middleton's article was included. In this paper, the author demonstrated that there is a correlation between sera levels of NETs and the severity of the disease. The data is shown in the bottom on page 8.

Comments: According to the modifications made to the text, we decided to review the conclusions. The changes to this section are described on page 14, lines 4 and 19.

Reviewer 3 Report

Bautista-Becerril and coauthors have submitted a review article on immunothrombosis in COVID-19 and the role of NETs .

There are several major concerns that need to be addressed before its publication.

Overall, the manuscript needs a thorough review of English formatting as in several instances the sentence is not well formatted to express its meaning.

Figure 1 is about the virus replication cycle which is well described in several other publications. Not sure why this figure is important in this article. The authors have mentioned about dysregulation of pathways like NFkB, AKT and JNK but those pathways have not discussed and not sure its relevant to the topic of this article. This figure can be deleted.

Table 1: Add additional 2 granules in the list.

The paragraph with NETs in viral infections: Delete this as the content is not related to the COVID-19 and moreover this a overview of the importance of NETs for other viral diseases.

A lot of redundancy between NETs in Immunothrombosis and Immunothrombosis, NETs and COVID sections. I think the authors should reorganize all those published observations in such a way under the heading of Immunothrombosis, NETs and COVID. The authors’ can start the section with the reports of thrombosis with cardiovascular disease as well as cerebrovascular disease and then can describe the importance of the major immune components (platelets, complements, cytokines and NETs) and their interaction.

Platelets: Review from Craig Jenne should be included (PMID: 27876233). If the platelets are not increased in the severe form of COVID disease, how the initial upregulation of platelets has an impact in leading the disease outcome is not clear. The authors should discuss this in detail.  Moreover, the last sentence in that section starting with “This finding could be beneficial to low-resource…” has no relevance with the conclusion of the data and can be deleted.

The paper will be beneficial to the reader if the role of NETs for both mild and severe form of the disease clearly defined and separated as a table of figure format. The role of cytokines and other biomolecules need to be included. NETs pathways are well known as a lot of study has been done on this mechanism. It will be important to show what happens during mild COVID infection and what differs in the mechanism from severe form of COVID infection. The present form of Figure 2 is not at all appealing.

Minor comments:

Page 1: last sentence: Please rewrite that sentence.

Page 2, line 2: Please rewrite the sentence where “severe” words have been used twice.

Page 3, epidemiology section, line 6th: By Feb 16th…. Please mention the year as without that its very confusing.

Several abbreviations have not explained when included for the first time.

Author Response

Title: “Immunothrombosis in COVID-19, implications of Neutrophil Extracellular Traps”

Reviewer 3. Responses.

Bautista-Becerril and coauthors have submitted a review article on immunothrombosis in COVID-19 and the role of NETs.

There are several major concerns that need to be addressed before its publication.

Point 1: Overall, the manuscript needs a thorough review of English formatting as in several instances the sentence is not well formatted to express its meaning.

Response 1: The English text was sent to the MDPI editorial for review. The language edition certificate is attached.

Point 2: Figure 1 is about the virus replication cycle which is well described in several other publications. Not sure why this figure is important in this article. The authors have mentioned about dysregulation of pathways like NFkB, AKT and JNK but those pathways have not been discussed and not sure it's relevant to the topic of this article. This figure can be deleted.

Response 2: Other publications describe the viral replication cycle very well, so based on your suggestion, we decided to remove figure 1.  In addition, since the paragraph "Receptor and signaling pathways" is not appropriate because signaling pathways are not discussed, we decided to change the title of the paragraph to "Mechanism of SARS-CoV-2 infection".

Point 3: Table 1: Add additional 2 granules in the list.

Response 3: Thanks for your suggestion. We added other granular contents to the list in table 1 (page 4).

Point 4: The paragraph with NETs in viral infections: Delete this as the content is not related to the COVID-19 and moreover this an overview of the importance of NETs for other viral diseases.

Response 4: We had considered as an important precedent, the role of NETs in other viral diseases. However, we consider we should center the paper only on the role of NETs in COVID-19, so we decided to remove this section from the article.

Point 5: A lot of redundancy between NETs in Immunothrombosis and Immunothrombosis, NETs and COVID sections. I think the authors should reorganize all those published observations in such a way under the heading of Immunothrombosis, NETs and COVID. The authors’ can start the section with the reports of thrombosis with cardiovascular disease as well as cerebrovascular disease and then can describe the importance of the major immune components (platelets, complements, cytokines and NETs) and their interaction.

Response 5: We appreciate your valuable feedback. We reorganized the suggested sections and decided to begin with the importance of biomolecules and cellular elements (platelets, complement and cytokines) and their interaction with NETs and subsequently addressing the findings of NETs in thrombotic complications in patients with COVID-19. These sections are found in the "Biomolecules and cellular elements involved in the interaction with NETs" (page 4) and "Immunothrombosis, NETs and COVID-19" sections (page 11. Additional information related to complement systems is shown at the bottom of the page 5. We also included a paragraph in the Venous Thromboembolism section on page 13 line 16.

Point 6: Platelets: Review from Craig Jenne should be included (PMID: 27876233). If the platelets are not increased in the severe form of COVID disease, how the initial upregulation of platelets has an impact in leading the disease outcome is not clear. The authors should discuss this in detail.  Moreover, the last sentence in that section starting with “This finding could be beneficial to low-resource…” has no relevance with the conclusion of the data and can be deleted.

Response 6:  We reviewed Craig Jenne's article and discussed the suggested paragraph. Additional information related to platelet levels and disease severity is shown on pages 6, line 3, and in the bottom of page 7.  The last sentence was removed from the platelet section as it clearly has no relevance for the conclusion of the data.   

Point 7: The paper will be beneficial to the reader if the role of NETs for both mild and severe forms of the disease is clearly defined and separated as a table of figure format. The role of cytokines and other biomolecules need to be included. NETs pathways are well known as a lot of study has been done on this mechanism. It will be important to show what happens during mild COVID infection and what differs in the mechanism from severe form of COVID infection. The present form of Figure 2 is not at all appealing.

Response 7:  We agree with the reviewer and consider these suggestions to be pertinent. The role of cytokine storm is included in the “Biomolecules and cellular elements involved in the interaction with NETs " section.  Regarding the comments concerning the differences between mild and severe forms of the disease, we collected information on biomarkers in the disease and discussed the differences according to the level of severity. The role of biomolecules and the differences between severity levels can be found on page 10 in the biomarker section in COVID-19. The image was re-designed to make it more attractive (page 9).

Minor comments:

Point 1: Page 1: last sentence: Please rewrite that sentence.

Response 1: We rewrote the sentence because it was confusing. Page 1: last sentence

Point 2: Page 2, line 2: Please rewrite the sentence where “severe” words have been used twice.

Response 2: We rewrote the sentence because it was confusing. Page 1 last sentence

Point 3: Page 3, epidemiology section, line 6th: By Feb 16th…. Please mention the year as without that it's very confusing.

Response 3: We agree, the year was included. Page 3, Epidemiology section, line 6

Point 4: Several abbreviations have not explained when included for the first time.

Response 4: The text was revised again and the abbreviations were explained when they were first included.

Comments: According to the modifications made to the text, we decided to review the conclusions. The changes to this section are described on page 14, lines 4 and 19.

Round 2

Reviewer 1 Report

In the revised manuscript, the authors review the pathogenesis, biomarkers, and possible therapeutic strategies for thrombosis caused by Neutrophil/NETs in COVID-19. The reviewer acknowledges that the manuscript has been properly proofread and is sufficient for acceptance.

Minor point: In page 5, is the description, “NOD-3-like receptor protein (NLRP3),” correct?

Reviewer 3 Report

The authors have addressed all concerns and the revised review manuscript is much informative and presentable.